# Acetone Sensors Based on Al-Coated and Ni-Doped Copper Oxide Nanocrystalline Thin Films

**DOI:** 10.3390/s24206550

**Published:** 2024-10-11

**Authors:** Dinu Litra, Maxim Chiriac, Nicolai Ababii, Oleg Lupan

**Affiliations:** 1Center for Nanotechnology and Nanosensors, Technical University of Moldova, 168 Stefan cel Mare Av., MD-2004 Chisinau, Moldova; dinu.litra@mib.utm.md (D.L.); maxim.chiriac1@mib.utm.md (M.C.); nicolai.ababii@mib.utm.md (N.A.); 2Department of Microelectronics and Biomedical Engineering, Technical University of Moldova, 168 Stefan cel Mare Av., MD-2004 Chisinau, Moldova; 3Department of Physics, University of Central Florida, Orlando, FL 32816-2385, USA; 4Functional Nanomaterials, Department for Materials Science, Kiel University, Kaiserstr. 2, D-24143 Kiel, Germany

**Keywords:** acetone, sensor, nanostructures, oxide

## Abstract

Acetone detection is of significant importance in various industries, from cosmetics to pharmaceuticals, bioengineering, and paints. Sensor manufacturing involves the use of different semiconductor materials as well as different metals for doping and functionalization, allowing them to achieve advanced or unique properties in different sensor applications. In the healthcare field, these sensors play a crucial role in the non-invasive diagnosis of various diseases, offering a potential way to monitor metabolic conditions by analyzing respiration. This article presents the synthesis method, using chemical solutions and rapid thermal annealing technology, to obtain Al-functionalized and Ni-doped copper oxide (Al/CuO:Ni) nanostructured thin films for biosensors. The nanocrystalline thin films are subjected to a thorough characterization, with examination of the morphological properties by scanning electron microscopy (SEM), energy-dispersive X-ray spectroscopy (EDX), and X-ray diffraction (XRD) analysis. The results reveal notable changes in the surface morphology and structure following different treatments, providing insight into the mechanism of function and selectivity of these nanostructures for gases and volatile compounds. The study highlights the high selectivity of developed Al/CuO:Ni nanostructures towards acetone vapors at different concentrations from 1 ppm to 1000 ppm. Gas sensitivity is evaluated over a range of operating temperatures, indicating optimum performance at 300 °C and 350 °C with the maximum sensor signal (*S*) response obtained being 45% and 50%, respectively, to 50 ppm gas concentration. This work shows the high potential of developed technology for obtaining Al/CuO:Ni nanostructured thin films as next-generation materials for improving the sensitivity and selectivity of acetone sensors for practical applications as breath detectors in biomedical diagnostics, in particular for diabetes monitoring. It also emphasizes the importance of these sensors in ensuring industrial safety by preventing adverse health and environmental effects of exposure to acetone.

## 1. Introduction

The development of advanced gas sensors and biosensors plays a key role in addressing various industrial and environmental challenges. Among the countless gases of interest, the detection of acetone vapor is particularly important due to its wide variety of applications in areas such as the cosmetics industry [1], the pharmaceutical industry [2], the paints and varnishes industry [3], the production of plastics and synthetic fibers [4], the production of chemicals [5], healthcare, biosensing, and environmental monitoring [6,7] (see Figure 1).

Acetone sensors are developed using a variety of semiconductor materials, such as CuO [8], Fe_2_O_3_ [9,10], TiO_2_ [11,12], WO_3_ [13], and different combinations: NiO/SnO_2_ [14], CuO/Cu_2_O/Cu-Fe_2_O_3_/Fe [15], CuO/Cu_2_O [16,17], Al_2_O_3_/CuO [18], ZnO–Fe_3_O_4_ [19], and TiO_2_/CuO/Cu_2_O [20]. Each of these has characteristics that influence the sensitivity, selectivity, and stability of the sensor. These materials enable acetone vapor sensing with variable responses that adapt to specific application requirements, ranging from high sensitivity and fast response time to the ability to distinguish acetone in the presence of other volatile compounds and to operate under varying environmental conditions. Therefore, the selection of the appropriate material and combinations is a pivotal step in optimal sensor performance in the targeted applications.

In healthcare, accurate acetone vapor detection is essential for numerous medical applications, including non-invasive methods for monitoring and diagnosing diabetes [21,22,23]. Diabetes is a chronic condition characterized by the inability of the pancreas to produce adequate amounts of insulin or the body’s inefficiency to properly use the available insulin [24]. High levels of acetone vapor in the breath can serve as an indicator of certain metabolic disorders, being an invaluable biomarker for diseases such as diabetes [25]. The acetone-responsive sensor for detecting diabetes is primarily a non-invasive method that can detect insignificant amounts of acetone vapor in exhaled air [26], and integrating this type of biosensor into a device in the future will allow minimizing invasive tests to visualize patients’ blood glucose levels [22]. Acetone vapor in exhaled air can occur in certain metabolic conditions, such as diabetic ketosis (DKA) or in people on a low-carbohydrate or ketogenic diet [27]. The mechanism of acetone in exhaled air primarily involves the breakdown of fatty acids in the body. Acetone is released into the blood and eventually exhaled through the lungs. As a result, people who experience high levels of ketosis, such as in diabetic ketoacidosis or during prolonged fasting, may experience acetone in the breath, leading to the characteristic fruity or sweet odor [25]. Figure 2 shows the proposed concept of acetone vapor detection in exhaled air by using an oxide based sensor structure, which will be portable and easy to handle. Worldwide, more than 500 million cases are expected to increase in 2030 due to sedentary lifestyles and an aging population [28]. Currently, the only way to monitor diabetes management on a daily basis is to use a solid-state device that works on the invasive principle of determining the patient’s blood glucose level [24]. In addition to detecting acetone to determine people with diabetes, acetone is a naturally occurring chemical compound found in the environment. Prolonged exposure to this chemical can cause negative effects on the body [29]. Exposure to acetone, in addition to poisoning through contact with detergents containing acetone, can damage the nervous, respiratory, cardiovascular, and endocrine systems due to its toxicity levels [30,31].

In industry, acetone is considered one of the most widely used ketones, and is involved in various chemical processes, for surface degreasing in organic chemistry and polymer chemistry [32] and in the manufacture of paints, varnishes, and lacquers [33], but at the same time, it is a chemical solution that is slightly flammable, even in vapors [34]. From all of the above, we can say that acetone is extremely useful in industry, but also extremely dangerous because it is a toxic chemical solution. Thus, solid-state sensors for acetone detection are today a necessity to prevent negative situations influencing the human body and the environment. Devices manufactured and used in the future will be based on metal oxide semiconductor sensors for the detection of acetone, which will increase the safety of people in the workplace, as well as the possibility of using the non-invasive method in the medical field and an efficient technology for its realization.

In this context, the use of Al/CuO:Ni nanostructures as a sensing material represents a cutting-edge approach for improving the sensitivity and selectivity of oxide-based acetone sensors. Zinc oxide (ZnO) nanostructures, known for their excellent semiconducting properties, are widely studied for gas sensing applications due to their high sensitivity and selectivity. Nickel doping in ZnO enhances these properties by affecting surface reactions and electrical resistivity, essential for improving gas sensing capabilities [35]. The incorporation of aluminum into the surface of nickel-doped copper oxide nanostructures introduces unique properties that make them promising candidates for efficient and reliable acetone sensing in medical diagnostics. The present work explores the novel design and fabrication of an acetone sensor based on Ni-doped copper oxide (CuO) nanostructured thin films coated with an ultra-thin Al layer followed by thermal annealing, with the aim of contributing to the advancement of gas sensing technologies with potential implications for extensive practical applications, especially in the field of medical diagnostics and metabolic health monitoring.

## 2. Materials and Methods

### 2.1. Sensor Manufacturing

To obtain the Al/CuO:Ni nanostructures, the solution chemical synthesis (SCS) method was used, following previous research [34,36]. In the synthesis process, all the necessary steps to obtain the structures, such as the preparation of the solution and all the conditions to obtain crystallization on the substrate were followed. Initially, to obtain CuO:Ni nanostructures, the copper oxide complex solution was doped with 10 mg of Ni(NO_3_) × 6H_2_O from Alfa Aesar to obtain CuO:Ni nanostructures after SCS, 56.99 at.% for copper, 42.83 at.% for oxygen, and 0.18 at.% for nickel, demonstrated by EDX and XRD measurements. The nanostructures were deposited on a glass substrate and then subjected to rapid thermal annealing (RTA) in air at 600 °C for 30 s. On the substrate was deposited Au contacts in the form of a meander through a mask by thermal evaporation. This went through several technological operations, such as heat treatment and metallization. Subsequently, the obtained nanostructures were sputter-coated with a 3 nm aluminum thin film, resulting in nanostructured Al/CuO:Ni thin films. The growth of ultra-thin Al films via thermal evaporation in a vacuum was accomplished using a vacuum system VUP, with a deposition temperature set at 100 °C. Such technology was employed to ensure conformal and uniform nanolayers, particularly for high-aspect-ratio nanostructures. Pure Al (99,999%) served as the aluminum precursor, while 30% RH H_2_O was used to oxidize the deposition during subsequent RTA. Next, the samples were thermally annealed at 625 °C for 2 h to activate impurities on the surface and in the bulk of the oxide. This is a simple and cost-effective method that allows controlled growth of the nanostructured films required for a sensor. The heat treatment was achieved using RTA technology, which allows rapid processing of the sample at temperatures that allow the obtainment of different copper oxide phases, such as CuO, Cu_2_O, or CuO/Cu_2_O [37]. Another reason is the elimination of defects after SCS deposition in the obtained structures and formation of nanocrystallites [38]. The reason for selecting CuO doping with Ni is due to the possibility of changing the selectivity from ethanol vapor [39] to acetone vapor, as reported by other authors in different semiconducting oxides [35,40,41]. The Al coating was used due to the fact that it is a material that contributes to the stability over time of the sensors as well as to the immunity to relative humidity [18].

### 2.2. Sensor Characterization

In this section, tools and methods to investigate the properties of Al/CuO:Ni sensor structures are described. The morphological and structural properties were investigated with different methods, such as scanning electrode microscopy (SEM, Carl Zeiss, Carl Zeiss) with an accelerating voltage of 7 kV, energy-dispersive X-ray (EDX) analysis (Zeiss Gemini Ultra55 Plus, Oberkochen, Germany), and X-ray diffraction (XRD) analysis (Seifert 3000 TT device, Ahrensburg, Germany) operated at a voltage of 40 kV and a current of 40 mA.

To record the gas response of the studied nanostructures, a stable and highly accurate measuring device (Keithley 2400 source meter from Tektronix, Cleveland, OH, USA)) was used. The data from the source meter are transmitted to the computer, where they are processed by LabView software version 17.0f2 (from National Instruments, Austin, TX, USA) and displayed as real-time graphs and saved data files. The following formula was used to determine the sensor signal [42]:(1)$$S=\frac{R_{gas}-R_{air}}{R_{air}}*100\%$$
where *S* is the sensor signal, *R_gas_* is the electrical resistance during exposure of the sensor to gas, and *R_air_* is the electrical resistance during exposure to air [43,44]. Another formula that can be used to determine the sensor response is:(2)$$Sr=\frac{R_{gas}}{R_{air}}$$
where *Sr* is the sensor response, *R*_gas_ is the electrical resistance during exposure of the sensor to gas, and *R*_air_ is the electrical resistance during exposure to air.

## 3. Results and Discussion

### 3.1. Morphological Characterization

The morphological properties were studied using a scanning electron microscope (SEM) at 7 kV. Figure 3a–c shows the SEM images of CuO:Ni nanostructures before and after RTA. Figure 3a shows the SEM image of the oxide structures before the heat treatment, in which we can observe interpenetrated copper oxide grains of different sizes at the 400 nm scale. A non-uniform surface can be observed on the grains. Figure 3b shows CuO:Ni nanocrystallites forming thin films after RTA at a temperature of 600 °C for 30 s. In the figure, it can be seen that the granules form the same structure as in the previous case, but on the surface of the granules, it can be observed that the protrusions have turned into a rough structure with steep peaks due to the diffusion of nickel in the crystal lattice. As a result of these deformations and formation of nanocrystals, the surface area-to-volume ratio increases, which contributes significantly to the improvement in the sensor properties. Figure 3c shows the nanostructures obtained at a magnified scale of 100 nm. Figure 3d shows the SEM image of the Al/CuO:Ni nanostructure thin films at a scale of 2 µm. Figure 3d shows a non-uniform grainy surface of the copper oxide with varying dimensions of nanocrystallites forming a thin film. Figure 3e shows an SEM image of the Al/CuO:Ni nanostructures at a scale of 400 nm, from which granulation or nanocrystallites can be observed more clearly.

Comparing the SEM images in Figure 3 before and after functionalizing, the same nanostructures with non-uniform grain size can be observed, the only difference being the coating of a thin Al layer of 3 nm on the surface of the CuO:Ni nanostructures such that the surface of the nanogranules became smoother in Figure 3d,e due to coverage with the Al nanolayer followed by thermal annealing in air, as can be seen in comparison with Figure 3a–c. The aluminum concentration is so low that its presence is not visible in Figure 3d,e. The SEM images in Figure 3a–c were obtained before aluminum coating. The SEM images in Figure 3d,e were obtained after coating with an ultra-thin aluminum nanolayer of 3 nm, followed by thermal annealing at 625 °C for 2 h in air for the elimination of defects, formation of nanocrystallites, and measurements at different operating temperatures and gases.

### 3.2. EDX Characterization

EDX analysis was carried out to prove the stoichiometric composition of the copper oxide structures. A layered mapping image of the CuO:Ni nanostructure is shown in Appendix A and the composition images are shown in Appendix A. This analysis was performed at a 2.5 µm scale and showed a uniform distribution of the Ni-doped CuO nanostructure, and the concentrations of this nanostructured films were 56.99 at.% for copper (Appendix A), 42.83 at.% (Appendix A), for oxygen and 0.18 at.% for nickel (Appendix A). As can be seen from the EDX analysis, Al is absent, the reason being that the measurements were performed before the deposition of the Al layer.

### 3.3. XRD Characterization

X-ray diffraction (XRD) was selected to investigate the crystallinity of the CuO:Ni structure before and after rapid thermal annealing at 600 °C for 30 s. Figure 4a shows the XRD plots of CuO:Ni structures before and after rapid thermal annealing at 600 °C for 30 s in a 2θ angle range of 30 and 45 degrees. A shown in Figure 4a, before the rapid thermal annealing (curve 1), the Cu_2_O phase was detected with reflections at 2θ of 36.28° and 42.12°, assigned to the Miller (111) and (200) planes, respectively, consistent with pdf sheet #05-0667 of Cu_2_O (Cu). The main reason for performing the XRD analysis is that the obtained results can be correlated with the gas sensor performance to optimize the material properties. For example, different crystalline phases or grain sizes can lead to variations in sensitivity, response time, and selectivity according to our previous experimental observations. By performing a comprehensive XRD study, one can correlate structural characteristics with functional performance, facilitating the design of sensors with improved characteristics.

The Cu_2_O phase had a cubic structure, the cuprite type, of space group $T_{h}^{2}-Pn3\;\left[65\;W\right]$ or $O_{h}^{4}-Pn3\;\left[72\;P\right]$ with the following unit cell parameters: a = 4.27 Å at p ≈ 0 GPa and a = 4.18 Å at p ≈ 10 GPa [82 W] [45]. Likewise, before rapid thermal annealing (Figure 4a, curve 1) NiO was also tentatively detected at 2θ of 37.12° and 43.28°, assigned to the Miller planes (111) and (200), respectively, in accordance with the card pdf #47-1049. For the CuO:Ni structures with rapid thermal annealing at 600 °C for 30 s (Figure 4a, curve 2) it can be observed that in addition to the reflections (hkl) attributed to Cu_2_O and NiO, the CuO phase appears, demonstrating the existence of mixed crystal phases of CuO and Cu_2_O, which is in agreement with the results obtained previously [37,45]. Thus, the reflections from 2θ of 32.44°, 35.5°, and 38.56° correspond to the Miller planes (110), (−111)/(002) and (111), respectively, according to card pdf #89-2529 of CuO (tenorite). CuO has a crystal structure characterized by monoclinic symmetry, with space group C2/c [46] with lattice constants a = 4.683 Å, b = 3.428 Å, c = 5.129 Å, α = 90.0°, β = 99.3°, and γ = 90.0° [17,47].

Figure 4b shows the XRD pattern of the CuO:Ni structures after rapid thermal annealing at 600 °C for 30 s in the 2θ angle range of 30–80 degrees. In Figure 4b, the presence of the CuO phase can be observed with the detection of reflections at 2θ of 32.44°, 35.5°, 38.56°, 46.4°, 48.8°, 51.8°, 53.4°, 58.2°, 61.4°, 65.2°, 66.1°, 67.6°, 72.4°, 72.8°, and 74.9°, which correspond to the Miller planes (110), (−111)/(002), (111), (−112), (−202), (112), (020), (202), (−113), (022), (−311), (113), (311), (221), and (004), respectively. Reflections for the Cu_2_O phase were detected at 2θ of 36.28°, 42.12°, 52.6°, 61.1°, 73.8°, and 77.2°, which correspond to the Miller planes (111), (200), (211), (220), (311), and (222), respectively. The reflections at the 2θ angle of 37.12°, 43.28°, and 62.4° correspond to the Miller planes (111), (200), and (220), respectively, for NiO, which is a face-centered cubic phase [48]. The high-intensity reflections of the CuO/Cu_2_O mixed phase indicate in Figure 4b that the structures have high crystallinity after RTA in air.

### 3.4. Gas Sensing Properties

In this investigation, the response of the developed sensor to a wide range of gases was analyzed, revealing a pronounced tendency for acetone detection at different concentrations. Comprehensive tests were carried out over a wide range of operating temperatures, from room temperature to 350 °C. Maximum response occurred at temperatures of 300 °C and 350 °C. This preliminary research emphasizes the distinctive selectivity of the sensor for acetone and paves the way for further investigation of its potential applications, particularly in contexts where accurate and responsive acetone detection is essential.

The results of testing the Al/CuO:Ni nanostructure thin film-based sensor with several gases demonstrated their selectivity for acetone vapors, which can be seen in Figure 5. The sensing properties were tested in acetone, methane, ammonia, 2-propanol, and carbon dioxide with a concentration of 50 ppm at operating temperatures of RT (22 °C), 150 °C, 200 °C, 250 °C, 300 °C, and 350 °C. For higher operating temperatures, higher gas response values can be observed, demonstrating the influence of temperature on the detection mechanism. Methane, ammonia, 2-propanol, and carbon dioxide have practically no gas response, but according to the formula used (2), their value cannot be less than 1. The error bar has a value of about 10% according to the technical specifications of the measurement device. Setting the selected gas or VOC vapor concentration to 100 ppm was used to calculate the flow of gas mixed with air in the relationship:(3)$$C(ppm)=\frac{C_{1}*F_{gas}}{F_{tot}}$$
where *C* is the required concentration of gas, *C*_1_ is the initial concentration of the test gas, *F_gas_* is the gas flow, and *F_tot_* is the total flow of the gas–air mixture.

The obtained Al-functionalized and Ni-doped copper oxide nanocrystalline thin film-based sensors were investigated at different concentrations of acetone vapor at an operating temperature of 350 °C. The results obtained after the investigation of these nanostructures to acetone vapor with a concentration of 50 ppm, 100 ppm, 500 ppm, and 1000 ppm are shown in Figure 6b, where the corresponding sensor signals ≈ 50%, ≈58%, ≈67%, and ≈79% can be observed. It can be seen that as the vapor concentration increases, the percentage value of the gas response increases, but the recovery time also increases. Figure 6a in the supporting information shows the response of nanostructures at concentrations of 1 ppm, 5 ppm, and 10 ppm and the corresponding responses of ≈ 5%, ≈12%, and ≈17%, respectively. In Figure 6b, it can be seen the small increase in response at each concentration shows that it is close to saturation concentration. At lower concentrations in Figure 6a, the increase is more significant, and this is the more appropriate result in this case, as they are concentrations closer to those exhaled by people with diabetes.

Figure 7a shows the gas response for each value of acetone concentration from 1 ppm to 1000 ppm. For each tested concentration, there is a corresponding response. Figure 7b shows the dynamic response measured at an operating temperature OPT of 350 °C to 1, 10, and 100 ppm acetone vapor concentrations. As can be seen, the response values obtained are ≈5%, ≈17%, and ≈58% at acetone vapor concentrations of 1, 5, and 100 ppm, respectively. The response times were calculated as τ_r_ ≈ 1.32 s, τ_r_ ≈ 2.54 s, and τ_r_ ≈ 9.83 s, respectively, with recovery times of τ_d_ ≈ 2.28 s, τ_d_ ≈ 9.57 s, and τ_d_≈38.54 s, respectively. The nanostructures were also tested for a period of 242 days (Appendix A), where approximately the same gas response was observed, demonstrating the stability of the obtained nanostructures over time. Results previously reported by us showed a worse response in cases without Ni dopant and without Al [17,48,49], or even the absence of a response to acetone for Al [50] and Ni [35].

The obtained results were compared with the results of other authors, where positive aspects were observed in some cases, such as high sensitivity compared to other results presented by other authors. Comparison of the obtained experimental data with the results from the literature are presented in Table 1. For the transformation (%) Formula (1) was used.

### 3.5. Gas Sensing Mechanism

Gas detection mechanisms proposed for semiconducting metal oxides help to explain the detection of certain measured gases or vapors in the environment. There are different types of gas sensors, each utilizing different principles for detecting and quantifying gas concentration. The operation of metal oxide semiconductor gas sensors is based on the variation in electrical conductivity of materials in the presence of certain gases [29,54,55]. When the target gas interacts with the surface of the oxide semiconductor, it increases or decreases electrical conductivity, resulting in a measurable change in electrical properties [54,56]. In the given case, CuO:Ni nanostructures are used to explain the mechanism of the sensor, since it is a *p*-type behavior semiconducting oxide, which means that when the gas is applied, the electrical conductivity decreases due to reducing the conduction channel, and the electrical resistance increases, as described in other previous work [30,34,56]. This can also be explained by charge transfer modification due to doping and makes oxide more sensitive to acetone vapor. Namely, Ni doping in CuO/Cu_2_O can promote dissociation on the surface and enhance preferential ionization of acetone [57,58].

Following the analysis of the gas sensitivity results, SEM images and X-ray diffraction analysis, the following sensor mechanism was proposed based on the results obtained and reported by other authors [55]. For cases with an operating temperature higher than 150 °C, oxygen species will form on the surface of the *p*-type oxide semiconductor as follows [37]:(4)$$\frac{1}{2}O_{2}↔O^{-}+h^{+}$$

If operating temperatures are increased above 150 °C, specific oxygen is adsorbed on the surface of *p*-type copper oxide (see Figure 8a) as $O^{-}$, the adsorbed and ionized atomic oxygen species, and the inducing of $h^{+}$, the holes in the CuO nanostructures [37].

Thus, Equation (5) shows the chemical reaction on the surface of the nanostructures, namely, the interaction of the acetone molecule with the oxygen species on the surface, as a result of which the gas dissociates into carbon dioxide and water [37]. A schematic representation is shown in Figure 8b.
(5)$${CH}_{3}CO{CH}_{3}+{8O}^{-}+{8h}^{+}\;⇌{3CO}_{2(gas)}+{3H}_{2}O$$

Aluminum coating of nanostructured thin films of nickel-doped copper oxide (CuO), even at low concentrations, as in the present case, plays a significant role in improving the gas detection performance, especially for acetone detection. Al coating modifies the surface properties of the CuO nanostructures. The Al layer alters the surface chemistry by introducing new reactive sites, enhancing the adsorption of gas molecules [50]. This is especially beneficial for acetone detection, as the Al coating improves the material’s sensitivity by creating more active sites for acetone molecules to interact with. Aluminum, being a conductive material, can enhance the overall conductivity of the CuO nanostructured films. This improvement in electrical conductivity reduces the baseline resistance of the sensor, enabling a clearer and more distinguishable response when exposed to acetone gas.

## 4. Conclusions

In summary, this work investigated the synthesis of SCS and morphological, structural, and acetone sensing properties of Al/CuO:Ni nanostructure thin films. These nanostructures were obtained using a cost-effective method, namely solution chemical sintering (SCS) followed by thermal processing using energy-efficient rapid thermal annealing (RTA) technology. The morphological study of the Al-functionalized and Ni-doped copper oxide nanocrystalline thin films was carried out using scanning electron microscopy (SEM), and revealed changes in the surface texture following different treatments. Before the heat treatment, the structure had a non-uniform surface with grains of different sizes, after the heat treatment the surface of the grains became rough with more pronounced edges, which increased the surface-to-volume ratio and contributed to an improvement in sensor properties. The high-intensity XRD reflections of the CuO/Cu_2_O:Ni mixed phase indicate that the structures have high crystallinity after RTA at 600 °C in air. The study of the morphological structures allowed us to understand the formation mechanism of these nanostructures and their selectivity to gases and volatile compounds applied on their surface. The sensor investigations revealed a remarkably high selectivity of Al/CuO:Ni nanolayers for acetone, even at different concentrations. EDX analysis was performed to confirm the stoichiometric composition of copper oxide nanostructures and Ni doping. Ni doping in CuO/Cu_2_O can promote surface dissociation and enhance the preferential ionization of acetone, and thus Al/CuO:Ni nanolayers become better sensors for acetone vapor. The experiments involved testing the sample over a range of operating temperatures from room temperature to 350 °C, with the highest sensitivities observed at 300 °C and 350 °C being *S* ≈ 45% and 50%, respectively, to 50 ppm gas concentration. Beyond the laboratory environment, these nanostructures show significant potential for integration into wearable and portable devices. Their selective response to acetone, especially at different concentrations, positions them as valuable components for non-invasive diagnostic tools, as well as biosensors. Their integration into cost-effective technologies for non-invasive diagnostics could pave the way to affordable and efficient detection methods, contributing to improving quality of life and advancing diagnostic capabilities in a variety of applications.

## Figures and Tables

**Figure 1 sensors-24-06550-f001:**
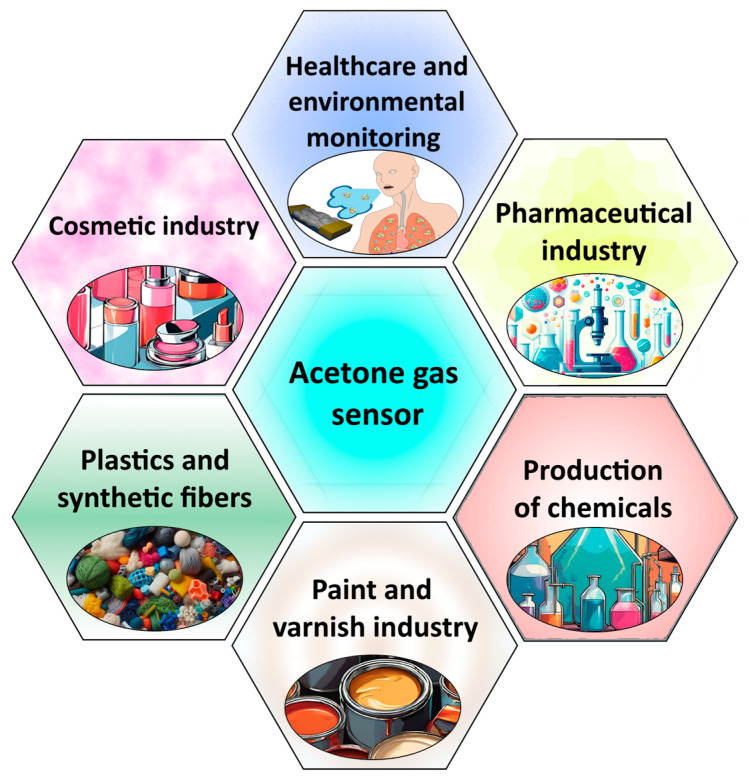
Graphical representation of the use of acetone sensors in different fields.

**Figure 2 sensors-24-06550-f002:**
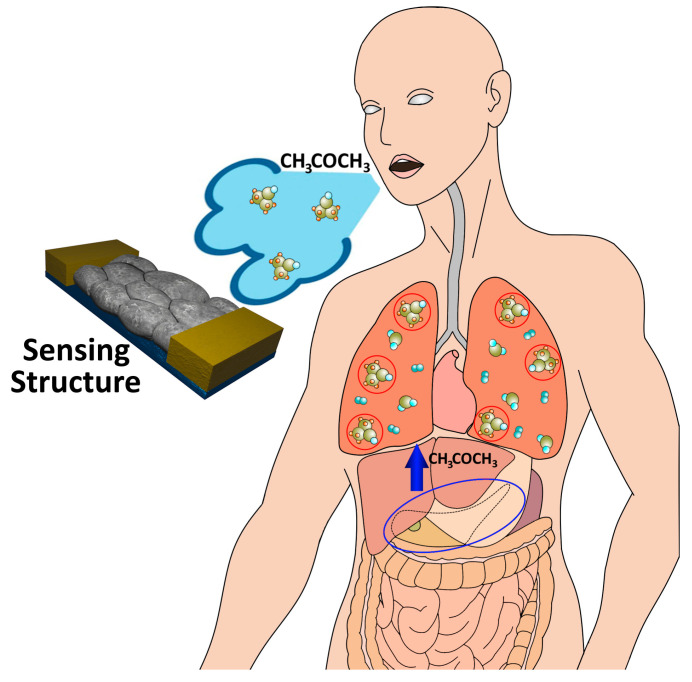
The proposed concept of acetone vapor detection in exhaled air using an oxide-based sensor structure.

**Figure 3 sensors-24-06550-f003:**
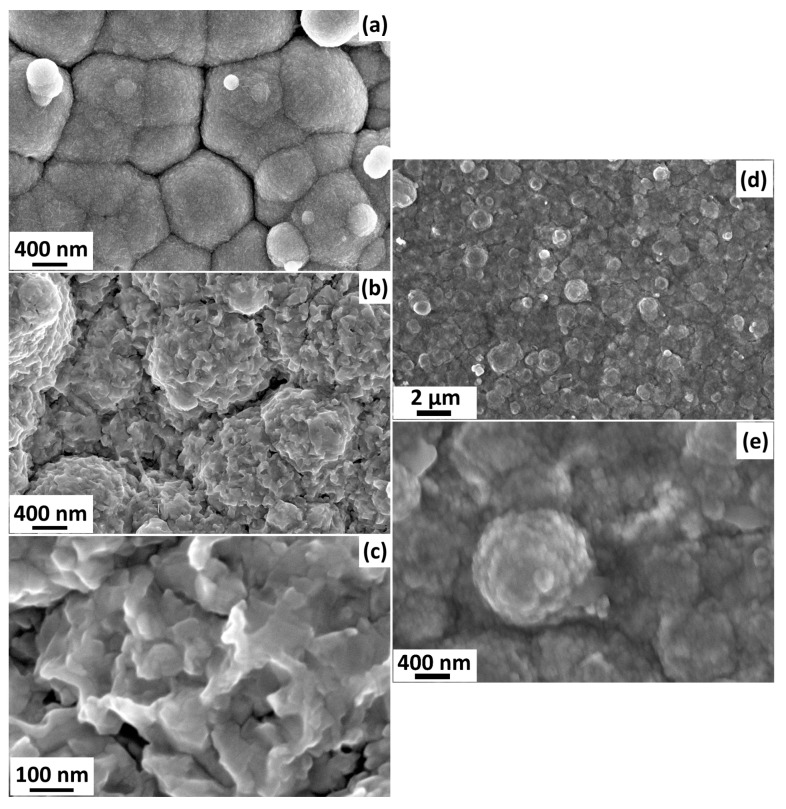
SEM images of CuO:Ni nanostructures: (**a**) before RTA treatment at a scale of 400 nm, (**b**) after RTA treatment (600 °C for 30 s in air) at a scale of 400 nm, and (**c**) after RTA treatment at a scale of 100 nm. SEM images after functionalization with Al and thermal annealing at a scale of (**d**) 2 μm and (**e**) 400 nm, respectively.

**Figure 4 sensors-24-06550-f004:**
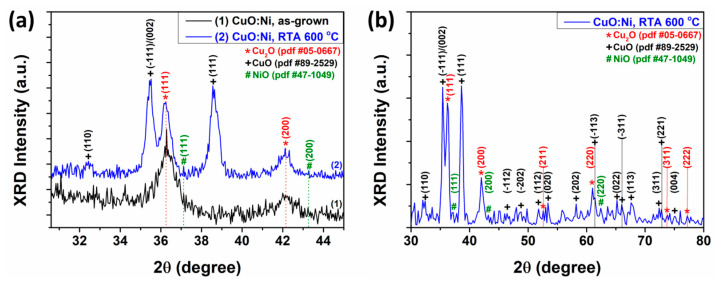
(**a**) XRD patterns in the range of 30–45 degrees of CuO:Ni thin films as grown and RTA-treated at 600 °C for 30 s in air and (**b**) XRD pattern in the range of 30–80 degrees of CuO:Ni after RTA treatment at 600 °C for 30 s in air.

**Figure 5 sensors-24-06550-f005:**
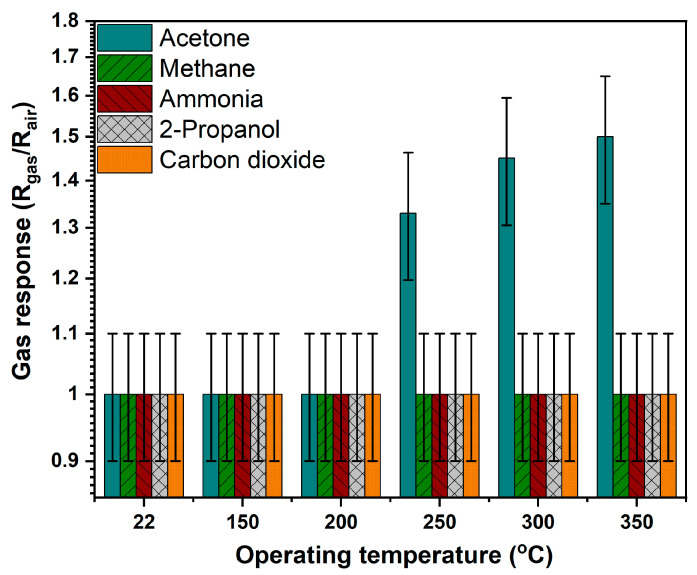
The response of Al-functionalized and Ni-doped copper oxide nanocrystalline thin films to acetone, methane, ammonia, 2-propanol, and carbon dioxide gases with a concentration of 50 ppm.

**Figure 6 sensors-24-06550-f006:**
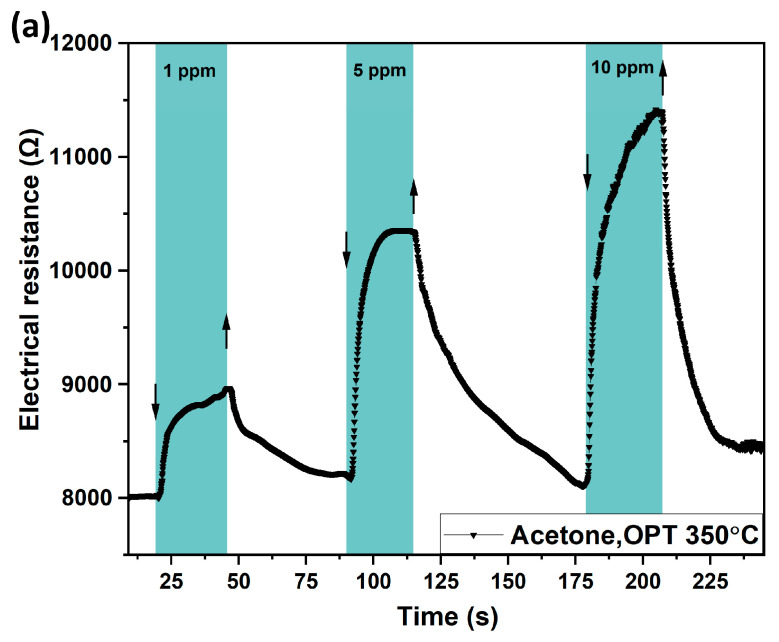
The dynamic response to acetone vapor of Al/CuO:Ni nanostructured film-based sensor at 350 °C at concentrations of: (**a**) 1 ppm, 5 ppm, and 10 ppm and (**b**) 50 ppm, 100 ppm, 500 ppm, and 1000 ppm.

**Figure 7 sensors-24-06550-f007:**
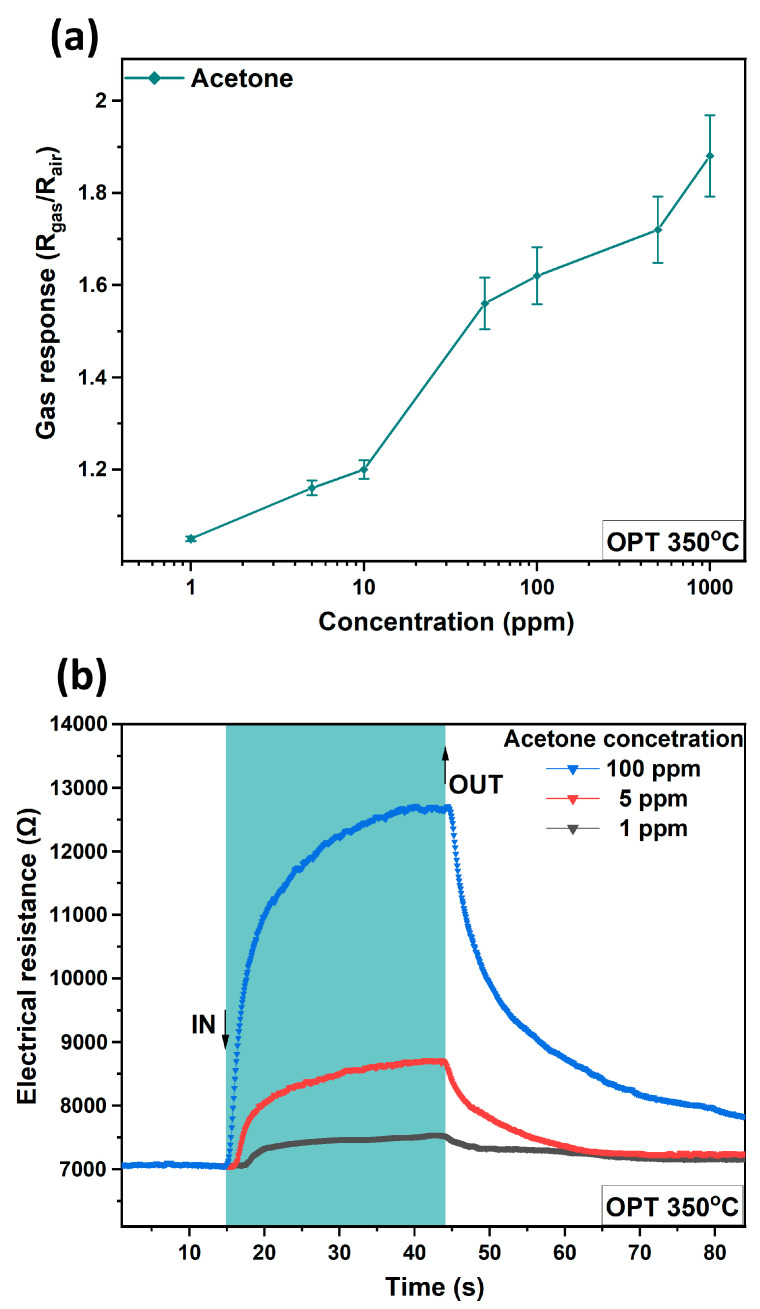
(**a**) Gas response as a function of acetone vapor concentrations for Al/CuO:Ni structure-based sensors at operating temperature of 350 °C. (**b**) Dynamic response at operating temperature of 350 °C of Al/CuO:Ni structure-based sensor to acetone vapor concentrations of 1 ppm, 5 ppm, and 100 ppm.

**Figure 8 sensors-24-06550-f008:**
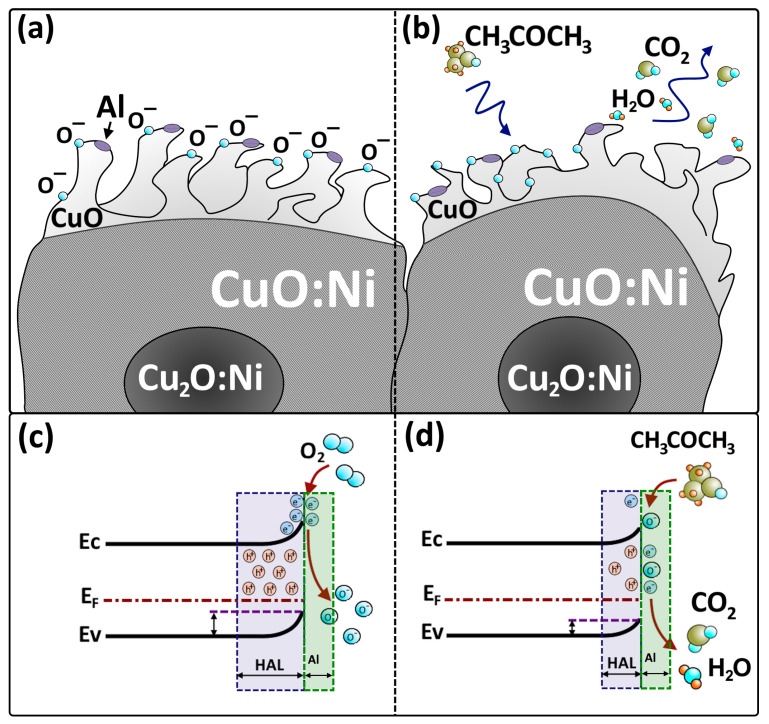
Schematic representation of the Al/CuO:Ni structure (**a**) in air and (**b**) in acetone vapors. Schematic representation of the corresponding energy bands (**c**) in air and (**d**) in acetone vapors.

**Table 1 sensors-24-06550-t001:** Comparison of the sensor’s response to acetone.

Material Type	Operating Temperature, (°C)	Gas Concentration, (ppm)	Response
CuO/Cu_2_/Cu-Fe_2_O_3_/Fe [15]	300	100	50%
CuO/Cu_2_O [17]	300	100	40%
Al_2_O_3_/CuO [18]	350	100	24%
ZnO [51]	300	1000–5000	27–42%
CuO [52]	350	100	23%
Pd-ZnO [53]	RT	1000	10%
**Al/CuO:Ni (this work)**	**350**	**100**	**58%**

^1^ Ra/Rg—electrical resistance ratio under exposure to air and target gas (acetone). For the transformation (%) Formula (1) was used.

## Data Availability

The data presented in this study are available on request from the corresponding author.

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
