# Peer review of "Acetone Sensors Based on Al-Coated and Ni-Doped Copper Oxide Nanocrystalline Thin Films"

_sensors, 2024, doi:10.3390/s24206550_

Round 1

Reviewer 1 Report

Comments and Suggestions for Authors

Reviewer’s Comments

The authors synthesized an Al-coated, Ni-doped CuO sensor, and the sensor demonstrated excellent acetone response and selectivity. However, the overall narrative of the manuscript needs to be more sophisticated before it can be accepted by Sensors.

Questions and Comments

1.      Title: “Al-functionalized” needs to be a more specific explanation, for example, Al-coated or something similar.

2.      Regarding “S = 45%” at line 30, you can’t use the abbreviated term without an explanation.

3.      In lines 118-119, please provide more details on how to synthesize the material in the text.

4.      In lines 121-122, please add sentences explaining how to prepare the sensor. Did you use an interdigitated electrode?

5.      In lines 191-196, I think it would be better to add the results related to Al.

6.      In lines 199-203, you don’t need to explain the principles of XRD measurement. Instead, please discuss the state of Al and explain why no peaks related to Al components were observed.

7.      In lines 253-255, I would like to see the results for the sensor using CuO:Ni without Al coating.

8.      For Figure 9, the effect of Al coating should be discussed.

Author Response

Dear Editor,

Sensors,

Thank you very much for reviewing of our manuscript.

We appreciate the positive and constructive comments of the Referees.

We take into account the comments and revised according to all reviewers' comments, and the itemized response to each reviewer’s comments is presented below.

Many thanks for your suggestions.

Reviewer's comments on the paper and the responses:

Comments and Suggestions for Authors

  1. Reviewer’s Comments

Comments: The authors synthesized an Al-coated, Ni-doped CuO sensor, and the sensor demonstrated excellent acetone response and selectivity. However, the overall narrative of the manuscript needs to be more sophisticated before it can be accepted by Sensors.

Questions and Comments

Comments 1.      Title: “Al-functionalized” needs to be a more specific explanation, for example, Al-coated or something similar.

Response 1: Thanks for the comment. The title has been modified as suggested

Comments 2.      Regarding “S = 45%” at line 30, you can’t use the abbreviated term without an explanation.

Response 2: Thanks for the comment. The abbreviated term has been modified as suggested. Please see revised draft for details.

Comments 3.      In lines 118-119, please provide more details on how to synthesize the material in the text.

Response 3: Thanks for the comment. In the synthesis process, the necessary steps were followed consecutively, such as the preparation of the solution, the respect of the reaction conditions and then the crystallization on the substrate.

Comments 4.      In lines 121-122, please add sentences explaining how to prepare the sensor. Did you use an interdigitated electrode?

Response 4: Thanks for the question. On the initial glass substrate was deposited Au contacts in the form of a meander through the meander mask, it went through several technological operations such as deposition, heat treatment and metallization. We made respective corrections in the manuscript. Please see revised draft for details.

Comments 5.      In lines 191-196, I think it would be better to add the results related to Al.

Response 5: Thanks for the comment. Unfortunately at the moment we don't have the possibility to do EDX analysis of the sample with Al since it is under detection limit of our device

Comments 6.      In lines 199-203, you don’t need to explain the principles of XRD measurement. Instead, please discuss the state of Al and explain why no peaks related to Al components were observed.

Response 6: Thanks for the comment. In the case of the XRD analysis the Al peaks are not observed because this analysis was performed before depositing the Al layer. But, samples covered with Al with 3nm thickness did not showed any peaks due to detection limit of the device, and extremely low ratio of materials

Comments 7.      In lines 253-255, I would like to see the results for the sensor using CuO:Ni without Al coating.

 Response 7: Thanks for the comment. Unfortunately the gas sensor investigations were carried out after the deposition of the Al layer. We have no possibility to repeat experiment at this stage. It will be reported in further works with more detailed comparison of different concentrations and ratios.

Comments 8.      For Figure 9, the effect of Al coating should be discussed.

Response 8: Thanks for the comment. The aluminum coating on the nickel-doped copper oxide (CuO) nanostructured thin films plays a significant role in enhancing the gas-sensing performance, particularly for acetone detection. Al coating modifies the surface properties of the CuO nanostructures. The Al layer alters the surface chemistry by introducing new reactive sites, enhancing the adsorption of gas molecules. This is especially beneficial for acetone detection, as the Al coating improves the material's sensitivity by creating more active sites for acetone molecules to interact with. Aluminum, being a conductive material, can enhance the overall conductivity of the CuO. nanostructured films. This improvement in electrical conductivity reduces the baseline resistance of the sensor, enabling a clearer and more distinguishable response when exposed to acetone gas.

Thank you very much for appreciation of our work. We considered all constructive comments from the reviewer, thus corrected the manuscript.

We corrected the manuscript and revised manuscript accordingly with reviewers’ comments.

The authors are grateful to the Editor for valuable time spent in processing and reviewing our research paper.

We deeply appreciate the reviewer for the critical review of our

manuscript with thoughtful and all constructive comments

With Best Regards,

Oleg Lupan                                                            

Professor, PhD.

Reviewer 2 Report

Comments and Suggestions for Authors

Before publication in Sensors MDPI, the work "Acetone sensors based on Al-functionalized and Ni-doped coper oxide nanocrystalline thin films" requires significant improvements.

The authors highlight the high sensitivity of Al/CuO:Ni sensors to different acetone concentrations, indicating the high potential of such materials for practical applications as breath detectors.

The following issues exist:

Instead of using formula (1) as percentage sensor response, please use S=Rgas/Rair which defines better the sensor response, not the sensitivity! Accordingly, at line 146, please change the definition of S from sensitivity to sensor signal.

How was the acetone concentration set to 50 ppm in the test gas? What was the reason why? Line 256. What about the error bars? Please specify for how many samples you have performed the gas sensing tests.

Figure 6 should plot the OY axis S=Rgas/Rair on a logarithmic scale to clearly show the sensor responses for the remaining gases.

In Figure 7 please use the classical representation of Electrical resistance versus time dependence instead of gas response. However, the tested acetone concentrations are extremely high, most likely occurring at industrial sites.

As long as the sensors do not attain equilibrium, the calculated response transients are subject to debate.

The same comments apply to figures 8 a and b. Instead of gas response (%) use the sensor signal S=Rgas/Rair and Electrical resistance on OY axis for Figure 8 b.

There is nothing new in the explanation regarding the Gas sensing mechanism (Section 3.5). Based on recent references in the field of gas sensing for acetone detection, I would suggest a more detailed approach.

There is nothing stated about the p-type behavior of CuO:Ni structure. Please include at least one reference to this aspect.

Since humidity was not considered in this study, I suggest to the authors that they rewrite the conclusions in Section 5 from lines 359 to 365. Their study is more fundamental and does not target a specific application in the field of non-invasive diagnostics.

Too many self-citations.

Author Response

Dear Editor,

Sensors,

Thank you very much for reviewing of our manuscript.

We appreciate the positive and constructive comments of the Referees.

We take into account the comments and revised according to all reviewers' comments, and the itemized response to each reviewer’s comments is presented below.

Many thanks for your suggestions.

Reviewer's comments on the paper and the responses:

Comments and Suggestions for Authors

  1. Comments and Suggestions for Authors

Comments : Before publication in Sensors MDPI, the work "Acetone sensors based on Al-functionalized and Ni-doped coper oxide nanocrystalline thin films" requires significant improvements.

The authors highlight the high sensitivity of Al/CuO:Ni sensors to different acetone concentrations, indicating the high potential of such materials for practical applications as breath detectors.

The following issues exist:

Instead of using formula (1) as percentage sensor response, please use S=Rgas/Rair which defines better the sensor response, not the sensitivity! Accordingly, at line 146, please change the definition of S from sensitivity to sensor signal.

Response 1: Thanks for the comment. The recommendations have been taken into account and the necessary changes have been made. Please see revised draft for details.

Comments 2: How was the acetone concentration set to 50 ppm in the test gas? What was the reason why? Line 256. What about the error bars? Please specify for how many samples you have performed the gas sensing tests.

Response 2: Thank you very much for pointing this. Setting the selected gas or VOC vapor concentration to 100 ppm was used to calculate the flow of gas mixed with air in the relationship:

(3)

where C is the required concentration of gas, C1 is the initial concentration of the test gas, Fgas is the gas flow and Ftot is the total flow of the gas-air mixture. A concentration as close as possible to that exhaled by people with certain diseases such as diabetes was chosen. The error margin is about 10%. Several such samples have been measured and the results are approximately the same. It was tested more than 5 samples sets.

Comments 3: Figure 6 should plot the OY axis S=Rgas/Rair on a logarithmic scale to clearly show the sensor responses for the remaining gases.

Response 3: Thanks for the comment. The recommendations have been taken into account and the necessary changes have been made.

Figure 6. The response of Al-functionalized and Ni-doped copper oxide nanocrystalline thin films to acetone, methane, ammonia, 2-propanol, carbon dioxide gases with a concentration of 50 ppm.

Comments 4: In Figure 7 please use the classical representation of Electrical resistance versus time dependence instead of gas response. However, the tested acetone concentrations are extremely high, most likely occurring at industrial sites.

Response 4: Thanks for the comment. The recommendations have been taken into account and the necessary changes have been made. The reason for testing at a higher concentration is the possibility of an alternative field of use such as industrial.

Comments 5: As long as the sensors do not attain equilibrium, the calculated response transients are subject to debate.

Response 5: Thanks for your observation. The sensor response stabilizes over time and does not change significantly with continued exposure to the gas as can be seen from the experimental data fig.7, 8, S2, S3. After longer waiting time, it recovers to initial values.

Comments 6: The same comments apply to figures 8 a and b. Instead of gas response (%) use the sensor signal S=Rgas/Rair and Electrical resistance on OY axis for Figure 8 b.

Response 6: Thanks for the comment. The recommendations have been taken into account and the necessary changes have been made.

Comments 7: There is nothing new in the explanation regarding the Gas sensing mechanism (Section 3.5). Based on recent references in the field of gas sensing for acetone detection, I would suggest a more detailed approach.

Response 7: Thanks for your observations. Section 3.5 has been modified as required.

Comments 8: There is nothing stated about the p-type behavior of CuO:Ni structure. Please include at least one reference to this aspect.

Response 8: Thanks for the comment. Unfortunately the gas sensor investigations were carried out after the deposition of the Al layer. Anyway, gas response showed p-type behavior of developed structures, similar to our previous works (P hys. Status Solidi RRL 10, No. 3, 260–266 (2016) / DOI 10.1002/pssr.201510414).

Comments 9: Since humidity was not considered in this study, I suggest to the authors that they rewrite the conclusions in Section 5 from lines 359 to 365. Their study is more fundamental and does not target a specific application in the field of non-invasive diagnostics.

Response 9: Thank you for your comment. It is known from our previous works that copper oxide-based sensor are stable to humidity. (Sens. Actuators B 153 (2011) 347–353.) Unfortunately the humidity investigations were not carried out the humidity in the chamber being 10%, but in future work this factor will be taken into account with more details.

Comments 10: Too many self-citations.

Response 10: Thank you for your comment. The recommendations have been taken into account and the necessary changes have been made to reduce the number of self-citations.

Thank you very much for appreciation of our work.

We corrected the manuscript and revised manuscript accordingly with reviewers’ comments.

The authors are grateful to the Editor for valuable time spent in processing and reviewing our research paper.

We deeply appreciate the reviewer for the critical review of our

manuscript with thoughtful and all constructive comments

With Best Regards,

Oleg Lupan                                                            

Professor, PhD.

Round 2

Reviewer 1 Report

Comments and Suggestions for Authors

Comments 6. In lines 208-213, you don’t need to explain the principles of XRD measurement. Please delete the following sentences.

Comments 8. For Figure 9, I suggested that the effect of Al coating should be discussed. But you didin't modify Fig. 8. Please modify the figure considering Al coating.

Author Response

Dear Editor,

Sensors,

Thank you very much for second review of our updated manuscript.

We appreciate the constructive comments of the Referees and its acceptance.

  1. Reviewer’s Comments

Questions and Comments

Comments 6. In lines 208-213, you don’t need to explain the principles of XRD measurement. Please delete the following sentences.

Thanks for the suggestion, it was a slip but now the suggested changes have been made. Please see revised draft for details.

Comments 8. For Figure 9, I suggested that the effect of Al coating should be discussed. But you didin't modify Fig. 8. Please modify the figure considering Al coating.

Thanks for the suggestion, it was a slip but now the suggested changes about fig.9 have been made. Although the concentration of aluminum on the surface is very low, when aluminum is exposed to air or other gases, it forms oxide molecules (AlOx) that create additional active sites on the surface. These sites contribute to the adsorption of gas molecules, thus increasing the sensitivity of the sensor. Please see revised draft for details.

Thank you very much for appreciation of our work.

We corrected the manuscript and revised manuscript accordingly with reviewers’ comments.

The authors are grateful to the Editor for valuable time spent in processing and reviewing our research paper.

We deeply appreciate the reviewer for the critical review of our manuscript with thoughtful and all constructive comments

With Best Regards,

Oleg Lupan                                                             

Professor, PhD.

Reviewer 2 Report

Comments and Suggestions for Authors

The authors have done a good job in reviewing their work.

Author Response

Dear Editor,

Sensors,

Thank you very much for second review of our updated manuscript.

We appreciate the constructive comments of the Referees and its acceptance.

  1. Reviewer’s Comments

The authors have done a good job in reviewing their work.

Thank you very much for the acceptance of our revised paper

All comments were properly addressed.

Thank you very much for appreciation of our work.

We corrected the manuscript and revised manuscript accordingly with reviewers’ comments.

The authors are grateful to the Editor for valuable time spent in processing and reviewing our research paper.

We deeply appreciate the reviewer for the critical review of our manuscript with thoughtful and all constructive comments

With Best Regards,

Oleg Lupan                                                             

Professor, PhD.